# OpenReview forum: "Training AI Co-Scientists Using Rubric Rewards"
_ICML.cc/2026/Conference — ICML 2026 regular_

### Official Review · Reviewer_NkY6 · 2026-03-06

**Soundness:** 2
**Presentation:** 2
**Significance:** 2
**Originality:** 3
**Overall Recommendation:** 3
**Confidence:** 3

**Summary:**

This paper studies how to train language models to generate better research plans for open-ended scientific goals. The authors build a scalable pipeline that extracts research goals, goal-specific rubrics, and reference solutions from scientific papers, then fine-tune a plan-generation model with RL using self-grading rubric rewards. The paper evaluates the approach on ML, medical, and arXiv-derived goals, and includes a human study with ML experts. The results suggest that rubric-guided RL improves plan quality over the initial model and shows some cross-domain generalization.

**Compliance With Llm Reviewing Policy:**

Affirmed.

**Final Justification:**

Thanks for the authors' response. I am not sure about the practical effectiveness of this method, so I am keeping my current rating.

**Key Questions For Authors:**

1. Can the authors provide stronger evidence that improvements in rubric scores and expert preferences translate into better downstream research outcomes, for example through limited execution-based validation on a subset of ML tasks?

2. Since the paper observes divergence between the self-grader and a stronger held-out grader during training, how stable is the method to the choice of reward model? Would training with a different self-grader or with periodically refreshed graders lead to similar conclusions?

3. Can the authors further disentangle whether the model is learning general research planning ability versus learning to better satisfy the extracted rubric format and scoring protocol? For example, would alternative evaluation settings without the same rubric structure yield similar gains?

4. The SFT result is interesting and surprising. Can the authors provide more analysis of why SFT degrades performance so sharply here, beyond style imitation, and whether this effect depends on the quality or granularity of the extracted reference solutions?

**Limitations:**

yes

**Strengths And Weaknesses:**

**Strength**

1. The paper addresses an important and underexplored problem: improving language models as AI co-scientists for open-ended research planning, where execution-based rewards are often unavailable or impractical.

2. The paper extracts research goals and rubrics from papers, uses rubric-based grading with privileged information, and trains with GRPO in a fully automated way. This is a meaningful attempt to scale supervision for difficult planning tasks.

3. The human study is a strong part of the paper. The reported 70% preference over the initial Qwen-3-30B model, as well as the comparison against Grok-4-Thinking, makes the empirical claims more convincing than relying only on LLM-as-a-judge results.

4. The paper includes useful ablations. In particular, the analysis of reward model capability, KL removal, specific versus generic rubrics, and SFT failure provides insight into what matters for this training recipe.

**Weakness**

1. The system is trained and mostly evaluated on rubric satisfaction and expert preference over plans, rather than on whether the proposed plans actually lead to better scientific outcomes when executed. The authors acknowledge this limitation, but it still leaves a gap between “better judged plans” and “better scientific assistance” in practice. This is especially important because the paper positions itself as a step toward AI co-scientists rather than only long-form writing models.

2. One of the more interesting results is that self-grading scores keep increasing while scores from a stronger held-out grader begin to diverge later in training. This is a valuable observation, but it also weakens confidence in the robustness of the training signal. The final checkpoint selection depends partly on this divergence behavior, suggesting that reward hacking remains a key problem.

3. The paper motivates this work as a scalable training recipe, but does not provide much discussion of inference cost, generation length, or cost-quality tradeoffs. Since the method relies on long-form planning and rubric-based grading, efficiency seems important for practical adoption.

---

> ### Author Rebuttal · Authors · 2026-03-30
>
> Thank you for recognizing the uniqueness of our scalable approach to this important setting, and for highlighting the strength of our human study and ablations. We now hope to address your remaining concerns.
>
> **W1 -- Why we use human expert preferences**
>
> We note that our work specifically targets the "AI Co-scientist" setting (e.g., Gottweis et al., 2025), where models assist human scientists in brainstorming—a use-case prominently highlighted in recent frontier model demonstrations (e.g. https://openai.com/index/gpt-5-medical-research/). For this application, our human study showing the trained 30B model's plans are preferred over the initial model (70%) and over Grok-4-Thinking (59.6%) offers strong evidence of our method's usefulness.
>
> **Q3 -- Evaluation validity**
>
> We carefully designed our evaluations to measure genuine planning capability rather than just rubric-following or long-form writing:
>
> 1. **Human Experts over 225 hours:** We explicitly evaluated research-specific criteria (soundness, experimental design, predicted outcomes; see Appendix C) rather than formatting. Qualitatively, experts found the trained models designed more precise, thorough experiments with better ablations.
> 2. **Alternative Evaluations (No Rubrics):** To explicitly address your question, we performed an LLM jury preference evaluation **without** the rubric structure (Appendix Table 7). The trained models were consistently preferred over the initial model across all tested families and sizes.
>
> **Q1 -- Manual execution**
>
> Manually executing a single generated plan would require weeks of human effort, so we couldn't do it. We acknowledge this practical limitation. During this study, we experimented with autonomous coding agents executing the plans, but found current models are not yet reliable enough. Once they are, they can autonomously execute plans for digital tasks like ML, letting humans evaluate the outcomes for a stronger evaluation. We will add this discussion to the revision.
>
> **W2/Q2: Reward modelling**
>
> Reward model overoptimization (Gao et al., 2023) is a fundamental challenge in RLHF and rubric-based RL. We explicitly analyze this effect in Figure 4, identifying the specific guidelines where Qwen3-30B self-grading diverges from stronger models. This analysis provides a roadmap for future work to perform targeted reward model finetuning.
>
> In Table 1, we demonstrate the stability of our methodology by training the 4B model using two different reward models (its own self-grading vs. the stronger 30B model). We see consistent improvements in both cases: the 30B RM pushed performance from an initial 12.0 to 23.3, while self-grading also led to a significant jump to 21.7. This confirms that the training signal is robust across reward models of different scales, though the ceiling for improvement naturally scales with the quality of the supervision.
>
> Regarding the suggestion of periodically refreshing the grader (e.g., via online human-in-the-loop annotations), we agree this is a highly promising direction for production-grade AI co-scientists. While beyond the current scope, our dataset is designed to support the development of such dynamic reward systems.
>
> **W3: Inference efficiency**
>
> We agree that efficiency is important for adoption. This is why we explicitly controlled for generation length throughout our methodology to ensure a fair, cost-neutral comparison:
>
> 1. **Structural Constraints:** All models are prompted to provide final plans within `<solution>` tags, constrained to ≤ 750 words.
> 2. **RL Length Penalty:** As described in L175-183 (L) and L181-185 (R), we incorporated a specific penalty for excessive length into the RL reward signal. This ensures that the RL-trained model maintains a similar plan length to the initial instruct baseline, resulting in improved plans at nearly identical inference costs.
> 3. **Thinking Length:** We do not observe any significant lengthening in the reasoning traces produced before the final solution.
>
> **Q4 SFT Degradation**
>
> Thanks for this suggestion. Degradation from SFT for long-form outputs has been noted in parallel work too (Wu et al., 2025b; Gunjal et al., 2025). To further quantify the criteria where SFT degrades, we measured its satisfaction of general guidelines (Appendix B.6) on the ML Validation set with Claude-4-Sonnet as the grader. We report the table in the response to Reviewer kUhe due to space constraints.
>
> We find that the SFT model indeed becomes more generic and fails to handle potential confounders or provide clear rationale. Notably, the reference solution scores far better across these guidelines. However, since it is a Llama-4-Maverick summary extracted from the original paper, it occasionally omits experimental details too. In contrast, Rubric-RL successfully rewards the presence of specific, necessary details.
>
> Thanks for your valuable suggestions, they helped make our work stronger!

---

> > ### Author Rebuttal · Reviewer_NkY6 · 2026-04-01
> >
> > Thank you for the detailed rebuttal. The additional no-rubric evaluation and the discussion of length control are helpful, and they strengthen the paper. However, I still think the main concern remains only partially addressed: the paper shows better rubric scores and stronger expert preference, but it is still unclear how directly these gains translate into better downstream scientific outcomes in practice. In addition, the observed divergence between the self-grader and stronger held-out graders suggests that reward hacking / reward-model dependence remains an important issue.

---

> > > ### Author Response · Authors · 2026-04-01
> > >
> > > Thank you for your quick response and suggestions that have helped improved our paper.
> > >
> > > We agree that the long-term goal of this research direction is building AI that creates improved downstream scientific outcomes. Our work can be considered an important step in this direction showing how training data can be scaled for the Co-scientist use-case, without needing to limit to tasks that can be performed in end-to-end executable sandboxes. Our recipe enables contributions toward AI Co-scientists at an academic scale, especially with the datasets we will release.
> > >
> > > We also asked the experts to compare the trained and reference models based on "Predicted Outcomes" of the study, with the guideline:
> > >
> > > > Which plan, if implemented to the word, do you think would lead to better research outcomes for the stated research goal?
> > >
> > > Experts preferred our finetuned 30B model's plans compared to the initial model for 72.5% research goals (Figure 2), and 62.4% over the much larger frontier model Grok-4-Thinking (Table 4).
> > >
> > > ## On reward-model dependence / "hacking"
> > >
> > > We would like to share a few points from the reward modelling literature which might help clarify that reward model overoptimization occurs for every proxy reward (related to Goodhart's law), and always depends on the reward model used to generate them.
> > >
> > > **Evidence from literature:**
> > >
> > > There has been a long line of work studying the dependence between reward models and overoptimization for language model post-training. Most recently, [Zhang et al. (ICLR 2026)](https://arxiv.org/abs/2509.21500)  show reward model overoptimization commonly occurs in our setting of Rubric-RL. We first quote a few lines from their related work section:
> > >
> > > > Reward over-optimization. Gao et al. (2023) highlighted the issue of reward over-optimization for both best-of-n sampling and reinforcement learning when using preference-based reward models ... this phenomenon has since been repeatedly observed in empirical studies (Bai et al.,2022; Moskovitz et al., 2023; Perez et al., 2023; Gui et al., 2024; Wang et al., 2024)
> > > >
> > >
> > > Indeed, [Gao et al. ICML 2023](https://arxiv.org/abs/2210.10760) showed this is a fundamental property of using proxy rewards. They show this occurs even with Best-of-N sampling which involves no optimization, so this is not related to stability of the optimization landscape. [Rafailov et al. (NeurIPS 2024)](https://arxiv.org/abs/2406.02900) shows this also occurs for DPO based RLHF for language model alignment.
> > >
> > > Parallel rubric RL work like [Unified Post-Training via On-Policy-Trained Language Model as a
> > > Reward Model](https://arxiv.org/abs/2603.18886), [Reinforcement Learning with Evolving Rubrics for Deep Research](https://arxiv.org/abs/2511.19399) has attempted to update the reward models or rubrics to continue training for longer. We think this is a promising direction, and was studied extensively as a way to delay overoptimization known as Iterated RLHF. It is worth noting that  [Wolf et al. 2025](https://arxiv.org/abs/2505.18126) found that even Iterated RLHF only delays the overoptimization, and inevitably also runs into this issue (in their setting after four iterations of reward model updates).
> > >
> > > Overall, mitigating reward model overoptimization remains a long-standing open problem, and early stopping has widely been used as a cheap proxy when additional data cannot be collected for updating the reward model (in our case, expert annotations are quite expensive). If you are interested, [Reward Hacking in Reinforcement Learning
> > > (Lilian Weng Blog)](https://lilianweng.github.io/posts/2024-11-28-reward-hacking/) is an excellent read on this topic, and highlights:
> > >
> > > > research into practical mitigations, especially in the context of RLHF and LLMs, remains limited.
> > >
> > > **Our own evidence:**
> > >
> > > 1. In Section 4.1, we report results from a human evaluation done with Machine Learning experts. Any hacking of the reward model from the checkpoint reported in our results would have been caught by them. They strongly prefer the trained model's plan to the initial model, and even to the much larger Grok-4-Thinking.
> > >
> > > 2. Our nested rubric design with goal-specific + general criteria helps analyze why overoptimization occurs which can guide future work on rubric/RM iteration. Without our goal-specific rubrics, it occurs faster (Appendix Figure 10).
> > >
> > > We hope our clarifications and new results in the rebuttal improve your support for our work, even if partially!

---

### Official Review · Reviewer_Zux6 · 2026-03-11

**Soundness:** 3
**Presentation:** 4
**Significance:** 3
**Originality:** 2
**Overall Recommendation:** 4
**Confidence:** 5

**Summary:**

In this paper, the authors provide a scalable approach to training co-scientists that can support different research goals by extracting research goals and rubrics to grade research plans from scientific papers. The authors conduct a human study with ML experts to demonstrate the benefits of training via rubric-guided self-grading. Empirical results also indicate cross-domain generalization effects.

**Compliance With Llm Reviewing Policy:**

Affirmed.

**Final Justification:**

I think this is a interesting and solid paper that could be impactful in the field. Though I have some concerns around the empirical findings in the paper, I think the authors are stepping towards the right direction. I would be happy to see the paper accepted if the authors can reframe some of their contributions and provide more details regarding their results. Therefore, I would like to recommend a weak acceptance.

**Key Questions For Authors:**

- The main results of the paper come from Qwen3-30B, which are somewhat limited. How would this approach scale from smaller models to (even) larger models?
- The Cohen's Kappa for human consensus vs. the model jury was relatively low (0.297). How can you make sure that the models are learning to designing desired research plans that can be really useful in open-ended plans?
- The authors speculate that lower performance on ArXiv tasks may be due to the lack of peer-review filtering for some papers in that dataset. How sensitive is the methodology to the quality of the "expert" source material?

**Limitations:**

yes

**Strengths And Weaknesses:**

Strengths:

- Training AI co-scientist is important yet challenging, especially when it comes to supporting different goals of different researchers. The proposed approach in the paper is principled and scalable by leveraging existing scientific papers.
- The combination of human evaluation and automatic evaluation with LLMs provides strong empirical evidence showing that training with rubric rewards can bring significant benefits.
- The paper also provides rigorous experimental design and detailed ablation studies that are inspirational for future research.  The paper is also clearly written with transparent methodology and qualitative examples.

Weaknesses:

- Rubric rewards have been widely discussed in recent research. However, the authors seem not to give sufficient insights into the rubrics design. It remains a little bit unclear about the different effects of different rubrics in this domain.
- Both evaluations seem to be performed at a coarse level, which are predictable to some extent. It is not surprising that models can align with humans within the paradigm, but it would be beneficial to systematically characterize the evaluations where models don't align with humans and provide insights into potential approaches.
- To prevent reward hacking, the method relies heavily on prompt-based techniques, which may be vulnerable to manipulation. The authors admit that as training progresses, the plans can become more complex and include "superfluous details" that may be "fooling" the 30B grader while not actually improving quality for stronger models like Claude-4-Sonnet.

---

> ### Author Rebuttal · Authors · 2026-03-30
>
> Thank you for recognizing our work as providing a principled and scalable approach for a challenging and impactful task and “inspirational for future research”.
>
> **W1:**
>
> > … insights into the rubrics design… unclear about the effects of different rubrics…
> >
>
> Recent literature debates the trade-off between general rubrics and sample-specific ones. We demonstrate how to leverage *both* by nesting general guidelines within goal-specific requirements (Section 2.2, Fig. 1 Right). The grader must verify that the relevant section of the plan satisfies *all* general guidelines before marking a goal-specific rubric item as complete (see Figure 5 in Appendix for a concrete sample).
>
> Regarding their individual effects, Table 1 explicitly ablates these components. Removing either the general guidelines or the goal-specific rubrics severely degrades performance (scores down to ~20.0 from 29% when both are used), demonstrating that both levels of granularity are essential for training.
>
> **W3:**
>
> > To prevent reward hacking, the method relies heavily on prompt-based techniques…"fooling" the 30B grader…
> >
>
> A core objective of our work is demonstrating how to achieve genuine capability improvements *before* reward hacking occurs. We mitigate overoptimization using rigorous, structural approaches that go well beyond simple prompt engineering:
>
> 1. **Extensive Human Validation:** Our 225-hour expert study validates that training generates better plans for 70% of research goals compared to the initial model, and 59.6% compared to the much larger Grok-4-Thinking. This confirms our models are not just "fooling" the 30B grader; human experts prefer the trained outputs, which is a high bar for evidence in reward modeling literature.
> 2. **Structured Grading:** Instead of asking the grader to score directly, we equip it with goal-specific rubric items and ask it to check general guidelines nested within each. This makes grading less subjective and makes it easier to decompose plan quality during training. Ablating either component leads to a large performance drop (Table 1).
> 3. **Evolution during training:** Section 4.3 (Fig. 4) analyzes how plans evolve. Notably, we stop training *before* reward hacking begins by checking with a stronger model as a proxy for human grading. This approach has been effectively used before for RLHF preference tuning (see Gao et al. 2023). Crucially, the plans become complex and details superfluous only *after* this stopping point (Appendix E.1.3).
>
> **Q1:**
>
> > main results of the paper from Qwen3-30B…limited.
> >
>
> We clarify that our methodology is not limited to a single architecture. In Appendix B.2, we demonstrate that our approach consistently improves performance across multiple smaller models (Qwen3-4B, Gemma-3-4B, Llama-3.1-8B). While RL-training models larger than 30B is computationally prohibitive for this study, we note that full RL fine-tuning at the 30B-parameter scale already goes beyond the majority of papers on LLM RL.
>
> **W2 and Q2:**
>
> Evaluating research plans is subjective; a Cohen’s kappa of ~0.3 provides a significant aggregate signal despite per-sample noise. Our primary claims rest on human preference, for which the automated metric is a highly predictive proxy: when it suggested our model outperformed Grok-4-Thinking (31.4% vs 30.5% rubric satisfaction), follow-up human studies confirmed this (59.6% preference). Furthermore, our evaluations are not “coarse”; human annotators validated that 84% of our goal-specific rubric items (Appendix Fig 5) test necessary, fine-grained features of a quality plan.
>
> **New Analysis: Human-LLM alignment analysis**
>
> Following your suggestion, we analyzed LLM jury explanations for disagreement patterns:
>
> - **Decreased Agreement:** Keywords like *“theoretically”* (p=0.03) or *“novelty”* (p=0.02) spike disagreement, as LLMs struggle with empirical reasoning or bias soundness toward novelty.
> - **Increased Agreement:** Keywords like *“ablation”*, *“thorough”*, and *“metric”* (p<0.01) correlate with human alignment.
>
> We will include this analysis in the revision, as it provides a clear roadmap for collecting targeted data to better align future reward models.
>
> **Q3**
>
> > How sensitive is the methodology to the quality of the "expert" source material?
> >
>
> We clarify that our observation regarding training on ArXiv papers referred to slightly lower *relative improvements* compared to training on ML/Medical, not lower absolute performance. While high-quality data naturally benefits post-training, our method remains robust to source quality. As shown in Table 1, even when we completely remove our data filtering step, the trained Qwen3-4B model still achieves strong performance (scoring 28.8 without filtering vs. 29.7 with filtering, up from a 12.0 baseline). This indicates that the core mechanism of rubric-guided self-grading drives significant capability gains even when applied to noisier source material.

---

> > ### Author Rebuttal · Reviewer_Zux6 · 2026-04-02
> >
> > I would like to sincerely thank the authors for their rebuttal. While the authors have clarified many points regarding my concerns, I don't think this paper alone can't address all of them. I still believe this work contributes a valuable path towards training LLMs to be scientific agents in the future. I would appreciate it if the authors could re-frame some points in their revisions. I think the additional human-LLM alignment analysis will be a valuable contribution as well. What is your guideline to curate those keywords? How do you recruit the participants? What is the experiment process? I am happy to raise my score accordingly based on the responses.

---

> > > ### Author Response · Authors · 2026-04-03
> > >
> > > We are glad to hear that you find our work valuable for the emerging, important direction of AI co-scientists! We will reframe our paper to elaborate on our rubric design, and how we systematically avoid risks in reward modelling like reward hacking. We will also more clearly acknowledge limitations such as relatively low agreement scores due to the subjective, open-ended nature of the task.
> > >
> > > We will also add a more detailed LLM - human agreement/disagreement analysis to the revision, and thank you for this valuable suggestion! For the analysis we provide in the rebuttal, we first identified prominent keywords related to plan quality by manually inspecting LLM jury and human annotations and seeing which ones seem to occur more often in the case of agreement/disagreement with human grading. We then quantified the effect of the keywords based on how frequently they occur in agreement vs disagreement cases. We can include the effect of this full list of important keywords we shortlisted in our revision, and will also try to do more types of analyses on the annotations.
> > >
> > > For the human evaluation, we recruited a pool of 25 experts for the human study on ML tasks through a paid annotation service, who were mostly graduate students and senior practitioners. We also take the subtopic the authors chose for each paper on OpenReview and used these to match relevant experts to samples. Each expert spent 45 minutes per plan comparison on average, which included going through detailed guidelines. Our human evaluation process is quite transparent, we provide the exact guidelines we gave to human annotators in Appendix C. As described here, our study was in two parts, the first one focusing on preference evaluations across important criteria like soundness and predicted outcomes, followed by ratings of rubric quality and whether model plans satisfy each of the rubric items. We discuss results of both in Section 4.1, with the human evaluators consistently finding our training leads to improved models in both parts (preference, rubric) and the rubrics we create of high-quality. We will be releasing our datasets publicly making the study quite reproducible.
> > >
> > > Thank you for helping make our paper stronger and your positive review! We noticed your score reduced from 4 to 3 before we submitted our rebuttal, so we hope our clarifications and additional analysis can further improve your support for our work. We are happy to incorporate any other suggestions you may have for the final revision.

---

### Official Review · Reviewer_kUhe · 2026-03-12

**Soundness:** 2
**Presentation:** 3
**Significance:** 3
**Originality:** 2
**Overall Recommendation:** 4
**Confidence:** 5

**Summary:**

The paper proposes a GRPO-based RL framework for generating research plans. It utilizes a self-grading mechanism where a frozen model evaluates outputs against goal-specific rubrics extracted from scientific literature, optimizing for constraint satisfaction without relying on human annotation or execution environments.

**Compliance With Llm Reviewing Policy:**

Affirmed.

**Final Justification:**

While I appreciate the connection to broader RL phenomena, I am maintaining my score because the paper would greatly benefit from a deeper, task-specific analysis of the reward hacking dynamics, especially in this open-ended task.

**Key Questions For Authors:**

- Can the SFT degradation be explained mechanistically (e.g., via or data selection)? Is the model structurally ignoring constraints to mimic the unconditional prior of the reference solutions?

- How does the discrete reward structure handle conflicting objectives? Have the authors tested continuous reward mapping or dynamic weighting based on gradient conflict?

- What structural modifications to the reward formulation or GRPO KL-penalty could fundamentally resolve the "detail" vs. "feasibility" reward hacking, beyond early stopping?

**Limitations:**

This paper did not acknowledge that its mitigation strategies for reward hacking (early stopping) and its analysis of SFT degradation are strictly operational. A more rigorous contribution would offer fundamental algorithmic solutions to these RL dynamics rather than just logging their symptoms.

**Strengths And Weaknesses:**

**Strengths**

- Exploitation of the Generator-Verifier Gap: By providing the reward model with privileged access to instance-specific rubrics, the framework successfully bootstraps reasoning capabilities without requiring a strictly superior oracle model.

- Empirical Optimization Bottlenecks: Benchmarking SFT against RL provides a concrete empirical data point justifying GRPO for long-form reasoning tasks

**Weaknesses**

- Superficial SFT vs. RL Analysis: The paper reports SFT failure without providing mechanistic insight. Treating the well-known open-ended generalization gap as a mere empirical log (without analyzing mode collapse or style-mimicking) lacks scientific depth.

- Phenomenological RL Dynamics: Reward hacking is observed (the 30B RM diverges from Claude-4-Sonnet on "efficiency" after step 120) but mitigated via a naive early-stopping workaround, lacking any structural algorithmic solution.

- Naive Reward Aggregation: Mapping 17 competing objectives (10 rubrics + 7 guidelines) into a flat, unweighted fraction obscures inherent gradient conflicts (e.g., "detail" vs. "efficiency"). The framework fails to map discrete LLM evaluations into a truly "learnable" continuous reward landscape.

- Task Leakage Risk: Extracting both the "Research Goal" and "Reference Solution" from the exact same paper using the same model without quantifying semantic overlap risks the model exploiting retrieval-based paraphrasing rather than ab initio planning.

---

> ### Author Rebuttal · Authors · 2026-03-30
>
> Thank you for appreciating our core contribution of leveraging the generator-verifier gap to bootstrap reasoning capabilities via self-grading. Below, we address the questions regarding the SFT degradation and our RL framework
>
> **W1/Q1:**
>
> > Can the SFT degradation be explained…mimic the unconditional prior of the reference solutions
> >
>
> We provide more analysis of SFT failing to generalize in Appendix B.4.2, where indeed we find that while the validation loss improves significantly from training, the model mainly learns to mimic the style of the reference solution (e.g. starting with “To improve… <restates the problem>”). The content of the solution itself worsens, with less rubric items being satisfied.
>
> To further quantify along which criteria the SFT model degrades, we now measure its satisfaction of general guidelines (similar to Appendix B.6) on the ML Validation set with Claude 4 Sonnet as the grader. As shown in the table below, we find it worsens significantly in addressing all rubric item requirements, becomes more generic, does not handle potential confounders and provide clear rationale for the chosen setup.
>
> | **Model** | **Handles all criteria** | **Detailed, specific solution** | **No overlooked weaknesses** | **Well justified rationale** | **Cost and effort efficient** | **No ethical issues** | **Consistent with overall plan** |
> | --- | --- | --- | --- | --- | --- | --- | --- |
> | **Qwen3-4B-Instruct-2507** | 49.84 | 23.04 | 40.6 | 53.88 | 80.96 | 97.0 | 85.52 |
> | **Reference Solution** | 83.02 | 29.56 | 60.86 | 47.92 | 90.26 | 99.14 | 97.62 |
> | **Qwen3-4B-SFT** | 31.88 | 8.88 | 30.1 | 35.78 | 85.98 | 93.5 | 82.42 |
> | **Qwen3-4B-RL** | 64.08 | 55.42 | 51.84 | 62.48 | 81.0 | 97.92 | 88.38 |
>
> **W2/Q3:**
>
> > Reward hacking is…mitigated via a naive early-stopping workaround
> >
>
> and
>
> > structural modifications to the reward formulation…"detail" vs. "feasibility" reward hacking, beyond early stopping?
> >
>
> Using stronger evaluators (or humans) for early stopping is a well-established, practical mitigation for reward overoptimization (Gao et al., 2023). We show (L358-364 L) why feasibility goes down for the RL model in Figure 4: Qwen3-30B’s self-grading diverges from stronger models.
>
> Structurally, this can be fixed by either finetuning smaller reward models to align their feasibility judgments with humans (Whitehouse et al., 2025), or using larger reward models. For instance, Appendix Table 12 shows GPT-5-Thinking has strong agreement (70.9%) with human graders on this criterion.
>
> In this study, we chose to use self-grading instead of stronger RMs to avoid the confounder of distilling capability from a better model. This strict constraint isolates our core finding: document-grounded rubrics can act as privileged information, creating a generator-verifier gap that drives improvements.
>
> **W3/Q2:**
>
> > Mapping 17 competing objectives (10 rubrics + 7 guidelines) into a flat, unweighted fraction obscures inherent gradient conflicts…
> >
>
> We clarify that the 7 guidelines serve as a structured mechanism to evaluate each goal-specific rubric item. A rubric item is marked satisfied *only* if none of these 7 guidelines are violated. Thus, our grading structure is strictly nested rather than flat, as illustrated in Figure 1 (Right) and detailed in Section 2.2. In Appendix Figure 5 provides a concrete sample illustrating how these rubrics function in practice.
>
> Your suggestion to identify appropriate weights for rubrics [1], or to develop methods for learning from rubric feedback [2], highlights an interesting direction in parallel work. Our primary contribution is the scalable data curation methodology for the challenging, expertise-intensive task of research plan generation. We validate this approach with the strong baseline of unweighted rubric RL, rather than engineering rewards to maximize performance. Our datasets enable research on exploring better methods to train models at generating research plans, an emerging and important topic as AI is increasingly used by human researchers as a co-scientist.
>
> **W4:**
>
> > Extracting both the "Research Goal" and "Reference Solution" from the exact same paper using the same model without quantifying semantic overlap…
> >
>
> To clarify, the reference solution is only ever provided to the grader, not to the model generating the plan. Furthermore, while we naturally expect semantic overlap between a research goal and its corresponding reference solution, we believe this is an inherent and desirable property of a well-posed scientific problem.
>
> [1] Shen, W. F., Qiu, X., Whitehouse, C., Alazraki, L., Goel, S., Barbieri, F., Willi, T., Mathur, A., & Leontiadis, I. (2026). Rethinking rubric generation for improving LLM judge and reward modeling for open-ended tasks. arXiv preprint arXiv:2602.05125.
>
> [2] Buening, T. K., Hübotter, J., Pásztor, B., Shenfeld, I., Ramponi, G., & Krause, A. (2026). Aligning language models from user interactions. arXiv preprint arXiv:2603.12273.

---

> > ### Author Rebuttal · Reviewer_kUhe · 2026-04-03
> >
> > I thank the authors for the helpful clarifications regarding the nested reward structure and the SFT degradation analysis. These points are well taken.
> >
> > However, my core concern regarding the mitigation of reward hacking (W2/Q3) remains. While I understand the experimental choice to rely strictly on self-grading (and therefore avoiding stronger models), using early stopping to prevent the policy from diverging is an operational workaround rather than an algorithmic solution.
> >
> > If the model's feasibility score collapses after step 120, it indicates that the current discrete reward formulation fails to provide a stable optimization landscape. For a framework proposing RL for complex reasoning, structural resilience against reward exploitation is a critical requirement. Halting training before the inevitable exploitation occurs leaves the underlying instability of the reward design unresolved.
> >
> > While the empirical results up to step 120 are interesting, this structural limitation impacts the robustness of the method. My concerns are therefore partially resolved.

---

> > > ### Author Response · Authors · 2026-04-03
> > >
> > > Thanks for your engaged discussions and clearly articulating your concern about our discussion of reward model overoptimization in Section 4.3. We would like to share a few points from the reward modelling literature which might help clarify that:
> > >
> > > 1. Reward model overoptimization occurs for every proxy reward
> > >
> > > 2. Reward hacking is not related to the stability of the optimization landscape or a weakness unique to our reward formulation
> > >
> > > 3. Early stopping is the canonical mitigation across the literature, similar to how it is a widely applied mitigation for overfitting in traditional supervised training
> > >
> > > 4. Structural resilience against proxy reward exploitation is challenging (Goodhart's Law)
> > >
> > > **Evidence from literature:**
> > >
> > > It is well-documented that when optimizing a proxy reward, eventually the learnt policy starts worsening on the "gold" reward, which is related to Goodhart's Law. There has been a long line of work showing this phenomena for language model post-training. Most recently, [Zhang et al. (ICLR 2026)](https://arxiv.org/abs/2509.21500) show reward model overoptimization commonly occurs in our setting of Rubric-RL. We now quote a few lines from their related work section:
> > >
> > > > Reward over-optimization. Gao et al. (2023) highlighted the issue of reward over-optimization for both best-of-n sampling and reinforcement learning when using preference-based reward models ... this phenomenon has since been repeatedly observed in empirical studies (Bai et al.,2022; Moskovitz et al., 2023; Perez et al., 2023; Gui et al., 2024; Wang et al., 2024)
> > >
> > > Indeed, [Gao et al. ICML 2023](https://arxiv.org/abs/2210.10760) showed this is a fundamental property of using proxy rewards. They show this occurs even with Best-of-N sampling which involves no optimization, so this is not related to stability of the optimization landscape. [Rafailov et al. (NeurIPS 2024)](https://arxiv.org/abs/2406.02900) shows this also occurs for DPO based RLHF for language model alignment.
> > >
> > > The benefit of using proxy rewards is that they are more scalable than expensive "gold rewards" (in our case human evaluation). But more expensive rewards than the proxy can be measured occasionally for checkpoint selection and early stopping. This makes early stopping using a validation set a natural, and widely applied mitigation for this phenomena, similar to how it is commonly used to avoid overfitting in supervised training settings. It is used across popular reward modelling work, for e.g. [Yuan et al. (ICML 2024)](https://arxiv.org/abs/2401.10020) and [Tian et al. (Neurips 2024)](https://arxiv.org/abs/2404.12253). In fact, Gao et al. show how KL divergence constrained RL is also similar to early stopping, with the analytical unit being KL divergence instead of training steps. Eventually, overoptimization occurs even in KL constrained RL.
> > >
> > > Parallel rubric RL work like [Unified Post-Training via On-Policy-Trained Language Model as a
> > > Reward Model](https://arxiv.org/abs/2603.18886), [Reinforcement Learning with Evolving Rubrics for Deep Research](https://arxiv.org/abs/2511.19399) and Zhang et al. (2026) has attempted to update the reward models or rubrics to continue training for longer. We think this is a promising direction, and was studied extensively as a way to delay overoptimization known as Iterated RLHF. It is worth noting that [Wolf et al. 2025](https://arxiv.org/abs/2505.18126) found that even Iterated RLHF only delays the overoptimization, and inevitably also runs into this issue (in their setting after four iterations of reward model updates).
> > >
> > > Overall, mitigating reward model overoptimization remains a long-standing open problem, and early stopping has widely been used as a cheap proxy when additional data cannot be collected for updating the reward model (in our case, expert annotations are quite expensive). If you are interested, [Reward Hacking in Reinforcement Learning
> > > (Lilian Weng Blog)](https://lilianweng.github.io/posts/2024-11-28-reward-hacking/) is an excellent read on this topic, and highlights:
> > >
> > > > research into practical mitigations, especially in the context of RLHF and LLMs, remains limited.
> > >
> > > **Our own evidence:**
> > >
> > > 1. In Figure 4, the dashed line (reward model scores but on the validation set) keep increasing with more training, showing that the optimization landscape is stable.
> > >
> > > 2. Table 1 shows adding a KL penalty leads to lower scores when evaluated with the strong grader. We also observed similar overoptimization with KL penalty as shown in Appendix Figure 9.
> > >
> > > 3. Our nested rubric design with goal-specific + general criteria helps analyze why overoptimization occurs which can guide future work on rubric/RM iteration. Without specific rubrics, it occurs faster (Appendix Figure 10).
> > >
> > > We appreciate you bringing up the importance of contextualizing our reward overoptimization results. We hope this discussion, which we will include in our revision, helps understand the result better, and increases your support for our work.

---

### Official Review · Reviewer_TKQw · 2026-03-13

**Soundness:** 2
**Presentation:** 3
**Significance:** 2
**Originality:** 2
**Overall Recommendation:** 3
**Confidence:** 5

**Summary:**

The paper primarily trains the LLM via RL to generate better research plans. The reward is designed using the "rubric as a reward" scheme where the rubrics are extracted from an already published scientific paper. The training dataset also contains the research problem, again extracted from the research paper along with the reference solution. They then use GRPO to train the model using these extracted rubrics.

The paper also proposes a dataset ResearchPlanGen that covers problems and plans for ML problems NeurIPS/ICLR, Medical problems from PubMed and problems from ArXiv. This dataset is a genuine contribution to the community.

**Compliance With Llm Reviewing Policy:**

Affirmed.

**Final Justification:**

I have increased my rating because the study does certain experiments but I am not very satisfied with either the direction or the results. I have explained more in the official comment.

**Key Questions For Authors:**

1. I would suggest running an experiment where the LLM completes a full implementation cycle based on the generated plan on a small set of research questions and have it evaluated by experts and based on the performance gains compared to the baselines.
2. Another beneficial analysis would be to investigate the feasibility/clarity regression, this may mean the model is learning to satisfy rubric items at the expense of actionability.
3.  If time permits, an experiment to measure the novelty would make the paper really strong, this might be done by computing similarity between generated plans and existing solutions.
4. Compare against prompting baselines.
5. See weaknesses.

**Limitations:**

Yes

**Strengths And Weaknesses:**

Strengths:

1. The method of extraction of research problems and plans is scalable requiring no human annotation.
2. The ablations are thorough isolating each component.
3. The dataset, ResearchPlanGen, is a contribution to the community.

Weaknesses:
1. Although the paper shows that LLMs become better at producing research plans when trained (which in itself seems obvious). There is limited understanding of whether the actual "research ability" of the LLM improves via such plan generation. I am arguing this because research is very open ended and once the LLM sees a new/novel problem, there is no set plan that it needs to follow to solve the novel problem. It requires search in open-ended space which might be hurt by such strict/definitive plan generation. The paper does not provide evidence of the downstream research ability of the LLM on novel/unseen research tasks. The plans are never implemented. We also don't know if improvements in the actual research ability of LLMs is substantive due to it being able to better generate plans extracted from existing research papers. Is the "better" generated plan using the metrics given in the paper "actually" a good plan for research?
2.  Human-judge agreement with LLM jury is weak, the Cohen's kappa only 0.297 and Claude-4-Sonnet gets a kappa of 0.076 with humans.
3. The finetuned model is worse on "clear to execute" (43.7% win rate) and "feasibility" (36.4%), which are important
  criterias.
4. Rubrics encode the paper's own solution, so the model may learn to copy published paper structures rather than generate genuinely good and practically executable research plans.
5.  There is no novelty assessment, which means that a plan that replicates the original paper's approach scores perfectly. Whereas, in real world, a novel plan would generalize better.
6. There is no analysis on the thinking tokens, the model might be hacking the reward, which is quite possible in such settings where there is a circular dependency between the reward and solution, potentially leaking information about the solution.
7.  No comparison against prompting baselines (few-shot, self-critique, best-of-N with same grader).

Significance to the community:

The overall contribution of the work is moderate and is primarily a recipe paper. The data extraction pipeline and ResearchPlanGen are useful. But methodological novelty is limited given concurrent rubric-based RL work, and without the actual execution of the generated plans on novel problems, the downstream usefulness and is unclear.

---

> ### Author Rebuttal · Authors · 2026-03-30
>
> We are glad you appreciate the scalability of our approach, our ablations, and the datasets we contribute to the community which lay the foundation for research on training better AI co-scientists in domains where execution-based verification is infeasible.
>
> **W1, W5, Q3**:
>
> > The paper does not provide evidence of the downstream research ability of the LLM on novel/unseen research tasks…
> >
>
> and
>
> > There is no novelty assessment…
> >
>
> We did evaluate the trained models on recent ArXiv preprints from August and September 2025 which came out after the model’s weights were frozen and released. The model consistently improves on these unseen tasks without search tool access (Section 4.2), indicating they generalize rather than simply extracting solutions from existing papers. Furthermore, our ML test set is derived from recent NeurIPS/ICLR papers ensuring many tasks are new.
>
> We would like to clarify that automatically executable tasks and novelty were deliberately not the scope of the study (see Introduction L37-40 (R)) as these have been studied before. Instead, we study the AI Co-scientist setting (Gottweis et al. 2025) where a human researcher articulates an open-ended research goal in domains with no predefined sandbox like biology, and the model assists them in coming up with a sound research plan. For such a setting, human experts preferring the model’s generated plan is a strong signal, as they are the target audience.
>
> **W1 and W4**
>
> > there is no set plan that it needs to follow to solve the novel problem...
> >
>
> and
>
> > Rubrics encode the paper's own solution, so the model may learn to copy published paper structures…
> >
>
> We agree that mimicking a set plan could hurt the model's ability to search open-ended spaces. Indeed, Section 4.3 demonstrates that SFT, which directly encourages mimicking reference solutions, degrades performance.
>
> For this reason, our training deliberately measures plan quality using rubrics instead of “matching” with the reference solution (see L151-153 (L)), and you can see Figure 5 for an example. We validated with human experts that 84% of our rubric items test for *necessary conditions* of a good plan, rather than copying the original paper’s structures. A similar rubric-based approach has been successful in prior work for open-ended evaluation (Arora et al., 2025) and training (Gunjal et al. 2025).
>
> For our main evaluation, human experts are never provided with the original paper or its reference plan; they simply compare the trained model’s plan against the initial model’s and Grok 4's (detailed in Appendix C). As reported in Section 4.1, they find the trained models plans better than the initial for 70% research goals, and better than the much larger Grok 4 Thinking for 59.6%.
>
> **W6**
>
> > …dependency between the reward and solution, potentially leaking information about the solution.
> >
>
> Since our evaluations are on unseen tasks, and also involve human experts, any such leakage would not benefit our evaluations, and we re-checked to ensure this is not occurring.
>
> **W2**
> As you rightly point out, generating a research plan is an inherently open-ended task, leading to natural disagreements even among human experts. While we acknowledge in L229-233 (R) and the Limitations section that per-sample grading can be noisy due to this subjectivity, the aggregated results across the full evaluation set provide statistically significant, large-effect evidence that our training meaningfully improves plan quality.
>
> **W3, Q2, W6**
>
> > investigate the feasibility/clarity regression, this may mean the model is learning to satisfy rubric items at the expense of actionability.
> >
>
> We clarify that “cost and effort efficiency” is included in the reward model's general guidelines when checking each rubric item, so satisfying rubrics is not in conflict with actionability. In Section 4.3, we show feasibility degradation is an artifact of self-grading with Qwen3-30B, as its cost/effort judgments align poorly with stronger models like Claude 4 Sonnet. We avoided training with a stronger reward model in this study as it would confound our results by introducing distillation from a more capable model, even though it would have better penalized overly complicated plans (see Appendix Table 12).
>
> **W7, Q4**
>
> Following the Fig. 3 (a) setup, we compare our rubric RL finetuned Qwen3-30B **(scores 31.4 +- 0.6 at no extra inference cost)** to the inference-scaling methods you suggest:
>
> 1. Few-shot prompting, using GPT5-Thinking (scores 46.5) responses to 3 train set samples. **Scores 29.3 at 4x prefill, 1x sampling cost.**
> 2. Self-refine (https://arxiv.org/abs/2303.17651) **scores 26.6 at 5.3x prefill, 2.3x sampling cost.**
> 3. Self-aggregation (https://arxiv.org/pdf/2509.14234) of 8 responses, **scores 29.8 at 9x prefill, 9x sampling cost**
>
> **Rubric RL is better with p < 0.05, at much lower inference cost.** Note that such inference scaling methods are complementary to the training based method we study.

---

> > ### Author Rebuttal · Reviewer_TKQw · 2026-04-04
> >
> > W7, Q4 resolved
> >
> > For other concerns, I do not align with the authors that making plans that match with existing research plans is going to lead towards better autonomous discovery. The authors should show that the application of these plans towards solving a research problems does in fact lead to better results. In my opinion, there is a requirement of diverse exploration even at the planning stage and training a model on existing research plans would lead to it following only good safe plans. This would hurt the downstream performance.
> >
> > I am willing to raise my score if the concern is resolved.

---

> > > ### Author Response · Authors · 2026-04-04
> > >
> > > Thanks for the engaged discussion. We are happy to clarify the remaining concerns.
> > >
> > > **Setting and our improved results on downstream human expert evaluations**
> > >
> > > The goal of our work is not fully autonomous discovery, which typically proceeds in end-to-end executable sandboxes with pre-defined metrics. In contrast, much of novel science involves designing the experimental setup itself, to analyze and reduce uncertainty about interesting research questions. In our work, the scientist provides the AI a novel, open-ended research goal, and the AI has to assist scientists by proposing a sound research plan. The goal for the AI generated plan is soundness, not novelty. This setting was proposed in the recent seminal work ["Towards an AI Co-Scientist" (Gottweis et al. 2025)](https://arxiv.org/abs/2502.18864) who studied inference-time approaches to better [assist scientists](https://research.google/blog/accelerating-scientific-breakthroughs-with-an-ai-co-scientist/). For the first time, we study how to go beyond inference-scaling, and train the model weights to generate research plans that scientists prefer, when no predefined execution sandbox exists to hillclimb.
> > >
> > > Executing these research plans is equivalent to carrying out a research study, which can take months per sample. Thus, a faster feedback proxy is needed to make research on this important problem feasible, which is one of the core contributions of our work. We do ask experts to compare predicted outcomes from our trained vs reference model's plans (trained model preferred for 72.9% goals shown in Figure 2), as well as their suitability for assigning to a graduate student to carry out the plan as a proxy (trained model rated's plans rated 7.89 vs 7.31 for initial model as in L254). This shows that our training leads to co-scientists whose responses experts prefer, which is a promising sign for an AI system meant to assist experts.
> > >
> > > **We are not matching to existing plans**
> > >
> > > We would like to clarify: our model's generated plans are NOT graded to match existing research plans. Our main evaluation consists of providing human experts only the original research goal (not plan), and showing them both our trained model's generated plan, and a reference model's. The experts then carefully analyze both plans, and compare them along axes like Addresses Requirements, Soundness, Predicted Outcomes etc. with results reported in Figure 2 when comparing to the initial model, and full guidelines for the study that we provide to the experts in Appendix C. The goals we test on are from recent papers, with the original paper not provided / unseen for the model. In a sense, we are back-testing the model on how it would approach designing experiments for recently studied research questions.
> > >
> > > Even our secondary automated evaluation using rubrics does not match to the existing research plans. The rubrics consist of "necessary" requirements that *any* plan for the research goal should satisfy, as validated in the second part of our human study (see Appendix D.1.2 for extraction prompt). These are based on desiderata the authors (who presumably spent many months thinking about the research question) state in their paper, for e.g. "The methodology should prevent overfitting by adaptively determining when to cease modifications based on consecutive documentation versions." from the example we show in Figure 5. The model is free to generate alternate plans, and indeed does so without penalties (See Appendix E.1.1 for grading samples), with summaries of the original paper's sometimes scoring lower than different plans generated by strong models like GPT-5 (see Appendix E.1.2). We also perform automated preference evaluations without the rubrics. plans or any other information from the original paper reported in Appendix Table 7 which again show our training improves models.
> > >
> > > **We are not training on existing research plans**
> > >
> > > As illustrated in Figure 1, we use the above rubrics that grade *necessary* requirements and general research criteria any research plan must satisfy for training, on the train-set using our scalable data curation methodology (core contribution of this work). The rubric grading leads to a reward (fraction of rubric items satisfied + length/format constraint penalty) which is used to reinforce the policy's generated plans during training. In no part of this process are we instructing the grader to "match" to the existing plan (see grader prompt in Appendix D.2.1). In fact, we agree matching to existing plans would hurt downstream evaluations, which is exactly what we find with the comparison to supervised finetuning reported in L370-384, which trains to match a reference plan and performs much worse, consistent with your intuition! This is exactly why our rubrics-based approach is valuable!
> > >
> > > Overall, we do think we are aligned in our intuitions with you. We will clarify these points better in Section 3 (Methodology), and hope this resolves your concern.

---

### Decision · Program_Chairs · 2026-04-30

**Decision:**

Accept (regular)

**Comment:**

In this paper, the authors propose a new post-training recipe using rubric-as-a-reward for generating better research plans. The primary contribution is automatic extraction of research goals and rubrics from a large collection of research publications across conferences, arXiv and journals from various fields. For the ML domain, they provide a well-designed ML-expert study (100 goals from top NeurIPS/ICLR papers, ~225 annotator-hours), in which the finetuned Qwen3-30B plans are preferred over the initial model on 70% of goals, with the authors also honestly disclosing per-criterion regressions on feasibility and clear-to-execute. The reviewers agree that the ResearchPlanGen dataset and human study are strong contributions, while there is substantial debate (from TKQw and NkY6) on the novelty of the post-training approach and improvements claimed, primarily because it is particularly challenging to evaluate whether a plan translates to solving a research question. The absolute gains are modest: 3-5 pp across ArXiv subjects on average and ~10pp improvements only on ML and Medical, and in MATH research, for instance, there have been no significant improvements from the finetuned model. For these reasons, I am leaning towards weak accept.